# Quality Models for Artificial Intelligence Systems: Characteristic-Based Approach, Development and Application

**DOI:** 10.3390/s22134865

**Published:** 2022-06-27

**Authors:** Vyacheslav Kharchenko, Herman Fesenko, Oleg Illiashenko

**Affiliations:** Department of Computer Systems, Networks and Cybersecurity, National Aerospace University “KhAI”, 17, Chkalov Str., 61070 Kharkiv, Ukraine; v.kharchenko@csn.khai.edu (V.K.); h.fesenko@csn.khai.edu (H.F.)

**Keywords:** artificial intelligence, quality model, quality, characteristic

## Abstract

The factors complicating the specification of requirements for artificial intelligence systems (AIS) and their verification for the AIS creation and modernization are analyzed. The harmonization of definitions and building of a hierarchy of AIS characteristics for regulation of the development of techniques and tools for standardization, as well as evaluation and provision of requirements during the creation and implementation of AIS, is extremely important. The study aims to develop and demonstrate the use of quality models for artificial intelligence (AI), AI platform (AIP), and AIS based on the definition and ordering of characteristics. The principles of AI quality model development and its sequence are substantiated. Approaches to formulating definitions of AIS characteristics, methods of representation of dependencies, and hierarchies of characteristics are given. The definitions and harmonization options of hierarchical relations between 46 characteristics of AI and AIP are suggested. The quality models of AI, AIP, and AIS presented in analytical, tabular, and graph forms, are described. The so-called basic models with reduced sets of the most important characteristics are presented. Examples of AIS quality models for UAV video navigation systems and decision support systems for diagnosing diseases are described.

## 1. Introduction

### 1.1. Motivation

Household comfort, quality of life, and safety of people are factors which are becoming increasingly dependent on information technology. Among such technologies are the most complex and promising means of artificial intelligence (AI). Evidence of the growing dynamics of the implementation of AI systems (AIS) in various fields, as well as the intensity of development and research, is the rapid increase in the number of publications during 2018–2021 [1], accepted and developed standards and guides EU Commission [2,3], ISO/IEC [4,5,6,7,8,9,10], IEEE [11,12], NIST [13,14,15,16,17,18], OECD [19,20,21], and UNESCO [22].

In industrial systems, healthcare, transportation, weapons systems, etc., the impact of AI is becoming increasingly tangible and sustainable, and on the other hand, it is very controversial in terms of implementation. This is due to:The versatility and complexity of decisions taken in the development and application of systems in which AI tools are built;The variability of the physical and information environments which are not always defined by the parameters in which they operate. The number and extent of external influences, such as cyberattacks aimed at artificial intelligence and based on AI methods are growing and expanding;The accumulation of expert information and expansion of knowledge bases that can be used to improve the efficiency of these systems. The principle of human-centeredness in its creation and application must be balanced to reduce the risks of wrong decisions due to subjective reasons;They increase the importance of ethical and safety aspects during use. This factor is especially important and specific to AISs. According to [22] and other documents focusing on the humanitarian aspects, human dignity, personal and collective security, and well-being are valued in the development and implementation of AISs.

These conditions, in contrast to “traditional” systems, complicate the formulation of specifications and verification of compliance with the requirements for creation and modernization. In addition, the number and variety of AI and AISs traits that need to be considered are growing, especially, ethics, clarity, credibility, etc. [23,24,25]. In turn, the methods of evaluation are diversified, which should be based on a clear idea of the nature and interdependence of the characteristics of artificial intelligence.

It should be emphasized that the increase in the number of publications and standards is accompanied by a significant disturbance in the characteristics of AI, which, on the one hand, determines, and on the other hand is due to a certain contradiction of definitions. Therefore, it is extremely important to research to harmonize and hierarchy characteristics, which objectifies and simplifies the development of tools for standardization, evaluation, and provision of requirements for the creation and implementation of AI systems.

### 1.2. Aim, Objectives, and Structure

The study aims to develop a model of the quality of artificial intelligence and AI systems based on the definition and ordering of traits.

Objectives:Formulate the principles and justify the sequence of analysis and development of quality models of AI and SSI as an ordered sets of characteristics;Analyze and classify the characteristics, determine their relationship and principles of use of AI and AI systems, provided in known sources, including standards and guides developed by leading institutions;Propose quality models of AI, AI platforms (AIPs) as parts of AISs, and systems using different forms of representation for further use, primarily as an evaluation of individual characteristics and quality in general;Demonstrate the profiling of AI and AISs quality models for systems using artificial intelligence (drone video navigation system (UAV) and urological disease diagnosis system).

The article is structured as follows. Section 2 grounds the principles of developing quality models and their sequence. Section 3 provides an overview of the references that provide definitions and describe the hierarchical relationships between characteristics. The codification of definitions is proposed, and the characteristics are separated from the principles of development and application of artificial intelligence. Section 4 proposes approaches to formulate definitions of AI characteristics based on the analysis of existing ones and their harmonization considering different groups of references, the final table with definitions and classification of AI quality traits is formed. The different forms of representation of dependencies and hierarchies of characteristics are presented in Section 5, and then it is used to analyze the relevant hierarchies based on the analysis of key sources. Section 6 describes the quality models of artificial intelligence and AI platforms, presented in parentheses, tables, and graphs. The so-called basic models with reduced sets of characteristics are given because of their importance. Section 7 describes examples of quality models for AI systems. Lastly, Section 8 provides conclusions and describes areas for further research.

The main contribution of the research includes a set of streamlined AI, AIP, and AIS characteristics, and quality models making it possible to specify requirements to systems and assess them during development and application.

## 2. Approach to Development of AI Quality Models

### 2.1. Principles and Concept

#### 2.1.1. Quality as a Generated AI Characteristic

**The set of characteristics of artificial intelligence systems analyzed in the article is united by the concept of “quality”** similar to how it is usually performed for software, where there are stable quality models that have developed and improved over 50 years of evolution [26,27,28]. The concept of “quality” of AI, in our opinion, is an acceptable generalizing feature, even though in some works it is used as a partial feature of AI, or study considers the quality of artificial intelligence purely in the context of software quality [29].

The software quality context is indeed very important, but it should be used as an approach to form a more general AI quality model. In [30], a position similar to the position of the authors of this study on the importance of AI quality is formed, although this work narrows the content of quality somewhat, as the set of AI traits analyzed is limited. Therefore, we further use the concept of AI and AIS quality as a system-forming, top-level entity in the hierarchy of all characteristics according to the general interpretation of quality according to ISO 9001: 2015 as the extent to which a set of objects-specific characteristics (in this case AI) to which a set of inherent characteristics of an object meets requirements). It is a guarantee for the development of an orderly set of characteristics of artificial intelligence.

Importantly, the quality of AI and AI system should be distinguished from the quality of the product that is the result of the AIS application. For example, the quality of images should be distinguished from the quality of AI with which the image has been processed. Although, there is an effect of AI (AIS) quality on product quality.

#### 2.1.2. AI, AI Platforms, and AI Systems

The quality of the AI system consists of the quality of AI as a generalized but specific object and the quality of the software and hardware platform (called as AIP), through which AI is implemented. This study considers only those components of the quality (characteristics) of AIPs that should somehow consider the specificity of AI as opposed to standard quality characteristics of software and hardware. It should be added, as the analysis showed, that there are quality characteristics that are common to AI and AIP. In this case, we will refer them to the characteristics of the quality of AI, but also consider them at the system level. The characteristics (models) of AI and AIP quality are combined in a general quality model. It is needed to note that requirements to common AI and AIP characteristics are implemented at the two stages: firstly, at the development of the models/algorithms considering specific features of AI functionality and design, and, secondly, at the development of hardware and software by the use of methods considering these features.

In addition, the ISO25010 standard [31], which describes the software quality model, divides the characteristics into two sets, namely subsets of the “product quality” and “quality in use” characteristics. In [30] there is a description of a fragment of such a distribution. Therefore, it may be the second feature to classify AI quality traits for more detailed analysis.

#### 2.1.3. Characteristics as a Key Conception

**The key concept used in the study is “characteristic”—a component of quality**, which describes the different characteristics of the AI, AI platform, and AI system. The characteristic is the basis for formulating requirements for the AI system and its components by:Considering the relevant characteristics in the development of the system specification, i.e., inclusion in the list of requirements;The definition and choice of metrics according to which the value of the property is evaluated, namely scales and methods of measurement;Substantiation of the necessary “limits” of this property, i.e., requirements to the qualitative or quantitative level thereof defined by the corresponding metrics.

If the characteristic Ch1 depends on the characteristic Ch2, then we will call the sub-characteristic of the characteristic Ch1. Ch1 and Ch2 should be located at the upper and next lower levels of the hierarchy of the quality model, respectively. For each of the characteristics (sub-characteristics), metrics must be defined for their evaluation, as well as formulated requirements for the values of these characteristics, and develop a profile of requirements and their quality [32].

Quality characteristics form a matrix classification, the general view of which is presented in Figure 1. We will call it the quality characteristics classification card (QCCC). QCCC columns correspond to the set of characteristics of artificial intelligence (AI characteristics), platform (AIP characteristics), including their common part (AI&PC), and the rows correspond to the sets of product quality characteristics (QPC), quality in use (QUC) and many characteristics that include both types of quality (QPUC).

Thus, the concept of artificial intelligence assessment is thus to develop a quality model as a general and orderly set of characteristics of the actual artificial intelligence, the corresponding software and hardware platform, and the AI system as a whole. Consider the stages of its implementation in the construction of quality models.

### 2.2. The Order of Quality Models Construction

Figure 2 illustrates the sequence, intermediate, and final results of constructing quality models of AI, as well as the AI platform and AI system. The results are given in the right part of the figure.

The main phases are as follows:In the first stage, the set of characteristics and principles of use of AI is formed based on the analysis of references, their coding is performed and the corresponding table is developed (Table 1, Section 3.1);Then, the separation of principles and guidelines for the use of AI from its characteristics, the results of which are given in Table 2 (Section 3.2);The analysis and harmonization of characteristics definitions according to the presented approach is performed. The result is Table 3 (Section 4). So, at this point, the formation of both the list of characteristics and their definitions is complete;To organize the characteristics according to their dependency, existing hierarchies are analyzed using their descriptions using simple brackets. The results of the analysis are presented in Table 4 (Section 5.1 and Section 5.2);A partially formalized procedure for building a hierarchy of AI quality traits is being developed. Accordingly, hierarchies between brackets, table, and graph forms are represented (Section 6.1);Then the models of AI, AI systems, and AI platforms quality are presented in visual graph form. Table forms of AI and AIP quality are given in Table 5 and Table 6 (Section 6.2), respectively. In addition, basic models are built, which are part of the overall quality models (Section 6.3);Finally, examples of profiling of quality models for two UAV video navigation systems and decision support for the diagnosis of urological diseases are provided (Section 7.1 and Section 7.2).

## 3. Analysis of References Related to AI Characteristics and Principles Definitions

### 3.1. Forming and Codification of AI Characteristics

The order for solving this problem is as follows.

The forming of a set of characteristics related to or may be related to AI, its platforms, and systems is as follows: the entire reference base, formed based on selected publications, was analyzed, and selected 75 characteristics named or defined in these references (Table 1).Selected AI characteristics were coded and provided alphabetically. The codification is performed using three Latin letters, which ensures their uniqueness.All references to the definitions in Table 1 are provided in the two right-hand columns:
Column with an indication of the reference where the characteristic is mentioned “Referenced in” provides links to publications where the relevant AI characteristic is only mentioned but not defined;Column with an indication of the reference where the definition of the characteristic is given “Defined in” provides references to publications where the characteristics of AI or AI systems are defined in any verbal version.

During the analysis and harmonization of the definitions of these characteristics, all references are divided into three groups:(GS) normative documents—standards, guidelines, technical reports of leading institutes and organizations, namely ISO, UNESCO, NIST, etc., as well as professional dictionaries and guides. These documents define the relevant characteristics to which we refer and take;(GR) scientific publications—articles, monographs, posts, etc., which provide their definitions of characteristics, or are based on the definitions provided in the links of the first group;(GV) academic dictionaries are used to define characteristics in the general sense when their definitions are missing in the documents of the first and second groups, or when they need to be clarified due to inconsistencies in the definitions given in the references of these groups.

Consequently, for each of the attributes in Table 1, the link in the column «Reference where the definition of the characteristic is given» is provided in a separate line (from the first to the third line according to the groups: GS, GR, and GV). Table 1, thus, collected 75 different terms related to characteristics and systematized references to relevant, most representative sources. Then their substantive analysis is performed.

### 3.2. Analysis of AI Principles

In the first phase of the analysis of all terms collected in Table 1, those that are not real characteristics but can be attributed to the principles related to the development of requirements, creation, and use of AI systems were identified.

Principles are generalized attitudes (values) and views that underlie the development, evaluation, and application of artificial intelligence, AI platforms, and AI systems.

Based on this, a set of characteristics is formed, which directly determines the various components of quality in a broad sense, and which can be measurable. Therefore, the terms “quality” and “quality model” are used as generalizations by analogy with the quality models of software and information systems.

In the process of analyzing key references related to the principles and characteristics of AI [13,14,15,16,17,18,22], as well as other sources provided in Table 1, the following are identified:The boundary between principles and characteristics is difficult to find because principles can dissolve into properties, and characteristics can be generalized into principles. Some characteristics are even identified with the principles;Even between different principles (groups of principles) there is a significant intersection in their components, which are in fact characteristics;The key difference between the characteristics and the principles is the possibility of their measurement (evaluation). Therefore, considering, firstly, the priorities of engineering practice, we have referred to the characteristics of the relevant concepts when such a possibility exists, i.e., there are or can be proposed criteria (scales and assessment methods) for it.

These conclusions are illustrated in Table 2, which provides the results of analyzing two important documents [2,22] and suggestions for the delineation of principles and characteristics.

In addition, in many sources and some sectoral regulations, such as [113,114], the principles, characteristics, and relevant requirements are formulated for both the AI systems themselves and for the personnel who develop and are responsible for their use of AI systems. It should also be noted that there are significant characteristics and differences in the list and detail of principles and characteristics for different industries, namely defense systems, healthcare, law, and education [13,19,20,113]. There are strong developments on standards for AI and AI systems in such areas, especially made by the IEEE [11,12] and other institutions, which is the subject of a separate analysis.

Thus, according to the analysis of Table 1, which contains 75 terms, with the excluded terms in Table 2 (seven terms related to the principles), it is necessary to analyze and harmonize the definitions for the remaining (68) characteristics.

## 4. Harmonization of AI Characteristics Definitions

The analysis and harmonization of the definitions of the characteristics of AI and AI platforms were performed as follows.

The analysis of definitions considered that individual characteristics can be identical, i.e., those that have different names but the same essence. Of the subsets of such properties, only one remained for further use. For example, of the characteristics “governance” and “controllability”, which means “ability to be controlled”, the characteristic “controllability” was left as more general. Plus, under the characteristics “explicability” and “explainability”, which means “ability to be explained”, for further consideration the characteristic “explainability” is chosen.Some characteristics with insignificant differences were combined, and these differences were considered in the relevant definitions. For example, the characteristic “human oversight and determination” was absorbed by the characteristic “human oversight”, considering the characteristics of AI supervision suggested in the absorbed characteristic in the final definition.Several characteristics have been excluded because, in our opinion, they do not have specific features for AI and AI platforms, but are common to technical systems or their software and hardware. Such characteristics include, in particular, “confidence” and “compliance”.For characteristics relevant to AI and AI platforms, the definition is provided by: Repetition (citation) or insignificant adjustment of the definition of one of the documents, which is the most adequate and accurate, according to the authors (marked with the letter R—referred). The definition of the attribute “integrity” was given, for example, by [31];Harmonization of definitions based on definitions provided in various publications (marked with the letter H—harmonized). The essence of harmonization was to identify key terms and combine the essential components of different definitions of the characteristic being analyzed. The definition of the trait “integrity” was obtained, for example, by combining the essential components of the definitions of this trait proposed in [6,31,33];Definition provided by the authors in the absence or in their opinion unsatisfactory wording for the description in the available sources (marked with the letter A—author). Thus, for example, the definition of the characteristic “resiliency” was obtained.

The relationship with the characteristics of different groups (AI, AI platforms, or AI and AI platforms) is determined in the wording itself, and the relationship with the type of quality characteristic (product quality QP and quality in use QU) is determined by a separate column.

The results of the analysis and harmonization of AI characteristics definitions are given in Table 3. Thus, 46 characteristics were selected, of which:Thirty-two are the characteristics of AI (22—exclusively the characteristics of AI), 24—the characteristics of AI platforms (14—exclusively the characteristics of AI platforms), and 10—the characteristics of AI and AI platforms at the same time. This division is made based on the analysis of each of the characteristics. The common characteristics include the following: accuracy, diversity, integrity;Twenty-nine characteristics are attributed to the quality characteristics of AI as a product (of which 7 are exclusively product quality characteristics), 39—to the quality characteristics in use (of which 17 are exclusively such characteristics), 22 characteristics are both characteristics of both types;The definition of 6 characteristics was selected from the relevant sources without changes; the definition of 36 characteristics is harmonized, and the definition of 4 characteristics is provided by the authors.

The results of the analysis are presented by the classification map of quality characteristics (Figure 3) according to the template developed in Section 2.1 (Figure 1). In this map, the corresponding characteristics in each cell are given, considering two parameters according to the object of quality assessment and the type of quality characteristics.

Quality characteristics in the classification map of quality characteristics cells (Figure 2) are written alphabetically without considering their importance or interdependence. Therefore, the next step should be to determine the dependencies of the characteristics and build a model of AI quality.

## 5. Description and Analysis of AI Characteristics Hierarchies

### 5.1. Description of AI Characteristics Hierarchies

Given a large number of characteristics of AI, AI platforms, and AI systems (Table 3), we present a description of the hierarchies of characteristics, determine the presence, and hierarchy of relationships to organize, and provide ease of assessment. Let us analyze how in the well-known sources, firstly, such leading organizations as ISO, UNESCO, NIST, and the EU special commissions, form the corresponding dependencies between the characteristics of AI.

The relationship of dependence and hierarchy between characteristics can be described in the form of simple parenthesis. This form is a multiple representation based on a hierarchy of characteristics, which is as follows:(1)X ={A, B, C {D, E}, F, H {G, I, J}},
where A, B, C, F, H—sub-characteristics of characteristic X (or quality characteristics of AI in general);D, E—sub-characteristics of C;G, I, J—sub-characteristics of H.

According to this notation, Table 4 provides the results of the analysis of those references where the verbal description of the already coded in Table 1, Table 2 and Table 3 characteristics’ hierarchy is taken place.

For example, for [2], Expression (1) is as follows:TST ={HMA, HMO, RBS, SFT, PRV, DGV, TRP {TRC, EXP}, DVS, NDS, FRN, SWB, ACN {ADT, RDR}}
where HMA, HMO, RBS, SFT, PRV, DGV, TRP, DVS, NDS, FRN, SWB, ACN—sub-characteristics of TST;TRC, EXP—sub-characteristics of TRP;ADT, RDR—sub-characteristics of ACN.

This expression is obtained through verbal analysis of the content of the document and its structuring.

### 5.2. Analysis of AI Characteristics Hierarchies

The conclusions of the analysis of Table 4 are as follows:The number of references, where in some ways the ratios of dependence on properties are given, is rather small. Of the more than one hundred references analyzed in Table 1 and considered in Table 3, the number of distinctive hierarchies that can be identified is 15;The most commonly used characteristics with described their hierarchies are trustworthiness TST (6 sources), explainability EXP (5 sources), responsibility RSP (3 sources), and ethics ETH (2 sources);The maximum number of hierarchy levels is 3, for example in [2] the TST property depends on the sub-characteristics of NMA, NMO, ACN, and the sub-characteristics of TRP and ACN depend on TRC, EXP and ADT, and RDR, respectively;

There are many inconsistencies in the proposed hierarchies:Firstly, the composition of the sub-characteristics, which are attributed to the relevant characteristics. For example, the (functional) safety of SFTs is attributed in some sources to the trustworthiness TST [2], which in our opinion is appropriate, and in others [38]—to the ethics of ETH, which is incorrect enough given their purpose;Secondly, by assigning them to a certain level of the hierarchy. Some characteristics (e.g., explainability EXP) in some sources [36] are characteristics of the first level of the hierarchy, and in others [17]—the second level or even the third one [2];Thirdly, the interpretation of these characteristics in general, due to inconsistencies in their definitions.

The discrepancies identified are most likely due to the fact that:Different documents have different directions due to different target audiences and domains (technical, ethical, legal, etc.), and are therefore insufficiently consistent;The definition of many characteristics is not provided in sufficient detail and clearly;Documents were prepared in parallel for a relatively short period (2–3 years) for such a complex and important problem;

Authors of non-institutional publications have focused on solving partial problems, so hierarchical dependencies were either not considered at all or were fairly specific [115]. Therefore, it is necessary to build a correct and consistent hierarchy of characteristics based on definitions and analysis of the relationship of dependency.

## 6. Model of AIS Quality

### 6.1. The Order of Building and Forming of AI Systems Quality Models

Construction of the AI systems quality model is performed in the following order:

**Stage 1.** Dividing the set of properties of the AI systems $S_{ChAIS}$ (Table 3) on the set of characteristics of pure AI, i.e., those who have specific characteristics of artificial intelligence, and those who are characteristics of software and hardware platforms which implement $\mathrm{AI}-S_{ChAIP}$:$$S_{ChAIS}=S_{ChAI\;}\cup \;S_{ChAIP},$$
it also true that
$$S_{ChAI}\cap S_{ChAIP\;}\neq \emptyset ,$$
since there are common characteristics for AI and AI platform.

According to the results of the analysis given in Section 4, we have two subsets of $S_{ChAI}$, $S_{ChAIP}$ and subset $S_{ChAI-ChAIP}$, which consists of joint characteristics of AI and AI platform.

**Stage 2**. Building of hierarchy for the quality model $QS_{ChAI}$ based on the analysis of properties set $S_{ChAI}$. When building a hierarchy in these and subsequent stages, the following procedure is used:

Step 1. Each of the characteristics of $S_{ChAI}$ are compared with all others and choose those that are dependent on others, and are those on which all others are not dependent (the ratio of dependence is determined by experts). Such characteristics must be attributed to the first level of the hierarchy $S_{ChAI-1}$ (with power $m_{1}$);

Step 2. Characteristics not included in $S_{ChAI-1}$, that is, formed a set
$$S_{ChAI-2}=S_{ChAI}\;\backslash \;S_{ChAI-1}$$
are divided into $m_{1}$ subsets of $S_{ChAI-2i}$, which do not intersect and affect the corresponding characteristics of the set $S_{ChAI-1}$:$$S_{ChAI-2i}=\cup \;S_{ChAI-2i};i=\left\{1,2,\ldots ,m_{1}\right\},\;A\;i\neq j:\;S_{ChAI-2i}\;\cap \;S_{ChAI-2j}=\emptyset .$$

Step 3. Steps 1 and 2 are repeated for each of the subsets $S_{ChAI-2i}$ with power $m_{2i}$, which makes it possible to form the second and third levels of the hierarchy.

This procedure continues, in the case of more levels in the hierarchy.

**Stage 3.** Building of hierarchy for the quality model $QS_{ChAIP}$ based on the analysis of properties set $S_{ChAIP}$ according to the procedure described for the Stage 2.

Stage 4. Combining hierarchies of sets of attributes $S_{ChAI\;}\cup \;S_{ChAIP}$ into a general quality model $QS_{ChAIS}$.

Thus, quality models are presented in three forms:Parenthesis (analytical) form, already used in the analysis of hierarchies of characteristics described in well-known publications (Table 3);Tabular form, in which the columns specify the levels of dependence on the characteristics, and their number is equal to the number of levels of the hierarchy for depth, and the rows specify the characteristics and sub-characteristics according to their dependence on individual groups;Graph form, the most obvious and convenient for further use to assess the quality of AI. In the graph, the vertices correspond to the characteristics and sub-characteristics, and the edges correspond to the relationship between them.

### 6.2. Building of AI, AI Platform, and AI System Quality Models

#### 6.2.1. AI Quality Model $QS_{ChAI}$

According to the step-by-step procedure, we form a set of characteristics of the first level. It includes the following characteristics $S_{ChAI-1}=\left\{ETH,\;EXP,\;LFL,\;RSP,\;TST\right\}$ because they are the most commonly used and directly affect the quality of AI.

The set of characteristics of the second layer (sub characteristics) are:
for ETH: $S_{ChAI-21}=\left\{\mathrm{FRN},\;\mathrm{GRS},\;\mathrm{HMA},\;\mathrm{HMO},\;\mathrm{RDR}\right\}$;for EXP: $S_{ChAI-22}=\left\{\mathrm{ACN},\;\mathrm{CSL},\;\mathrm{CMT},\;\mathrm{CMH},\;\mathrm{TRP},\;\mathrm{INP},\;\mathrm{INR},\;\mathrm{VFB}\right\}$;for LFL, RSP: $S_{ChAI-23}=S_{ChAI-24}=\emptyset$;for TST: $S_{ChAI-25}=\left\{\mathrm{DVS},\;\mathrm{RSL},\;\mathrm{RBS},\;\mathrm{SFT},\;\mathrm{SCR},\;\mathrm{ACP},\;\mathrm{ACR}\right\}$.

Next are the characteristics of the third level:$$S_{ChAI-1}=\left\{\mathrm{BIS},\;\mathrm{NDS},\;\mathrm{TRC},\;\mathrm{SWB},\;\mathrm{PRV},\;\mathrm{ING},\;\mathrm{OBC}\right\}.$$

Thus, the parenthesis form of the model has the following representation:$$QS_{ChAI\;}=\{\mathrm{ETH}\;\left\{\mathrm{FRN}\;\left\{\mathrm{BIS},\;\mathrm{NDS}\right\},\;\mathrm{GRS},\;\mathrm{HMA},\;\mathrm{HMO},\;\mathrm{RDR}\right\},\;\mathrm{EXP}\;\{\mathrm{ACN},\;\mathrm{CSL},\;\mathrm{CMT},\;\mathrm{CMH},\;\mathrm{TRP}\;\left\{\mathrm{TRC}\right\},\;\mathrm{INP},\;\mathrm{INR},\;\mathrm{VFB}\},\;\mathrm{LFL},\;\mathrm{RSP},\;\mathrm{TST}\;\left\{\mathrm{DVS},\;\mathrm{RSL},\;\mathrm{RBS},\;\mathrm{SFT}\;\left\{\mathrm{SWB}\right\},\;\mathrm{SCR}\;\left\{\mathrm{PRV},\;\mathrm{ING},\;\mathrm{OBC}\right\}\;\mathrm{ACP},\;\mathrm{ACR}\right\}\}.$$

The tabular form of the model is presented in Table 5. The asterisks (*) in the table indicate the characteristics common to AI and AI platform.

The graph form of the model is shown in Figure 4.

#### 6.2.2. AI Platform Quality Model $QS_{ChAIP}$

The peculiarity of this model is that it combines many of the actual platform characteristics of $S_{ChAIP1}$, as well as characteristics of $S_{ChAIP2}$, which is also part of many AI quality characteristics. At the platform level, these characteristics should also be considered when assessing the quality of AI platform.

Generally, the AI platform quality model is formed from a set of properties of $S_{ChAIP1}$ by the same procedure as the previous AI model. The set of the first level consists of the following characteristics
$$S_{ChAIP1}=\left\{\mathrm{ACS},\;\mathrm{ADT},\;\mathrm{AVL},\;\mathrm{CNT},\;\mathrm{EFS},\;\mathrm{INF},\;\mathrm{RLB},\;\mathrm{MNT},\;\mathrm{SST},\;\mathrm{USB}\right\},$$
and the second level—according to the characteristics $S_{ChAIP2}=\left\{\mathrm{DGV},\;\mathrm{MTR},\;\mathrm{TRF},\;\mathrm{GRN}\right\}$.

Thus, the parenthesis view of the model is as follows:$$QS_{ChAIP}=\left\{\mathrm{ACS},\;\mathrm{ADT},\;\mathrm{AVL},\;\mathrm{CNT}\;\left\{\mathrm{DGV}\right\},\;\mathrm{EFS},\;\mathrm{INF},\;\mathrm{RLB}\;\left\{\mathrm{MTR}\right\},\;\mathrm{MNT}\;\left\{\mathrm{TRF}\right\},\;\mathrm{SST}\;\left\{\mathrm{GRN}\right\},\;\mathrm{USB}\right\}$$

It should be noted that the models $QS_{ChAIP}$ the following characteristics joint with $QS_{ChAI}$ are added:$$S_{ChAI-ChAIP}=\left\{\mathrm{VFB},\;\mathrm{DVS},\;\mathrm{RSL},\;\mathrm{RBS},\;\mathrm{SFT},\;\mathrm{SCR},\;\mathrm{ACR}\right\}.$$

The tabular form of the model is presented in Table 6. The asterisks (*) in the table indicate the characteristics common to AI and AI platform.

The graph form of the model is shown in Figure 5, where the additional vertex AIG combines the characteristics of the set $S_{ChAI-ChAIP}$.

#### 6.2.3. AI System Quality Model $QS_{ChAIS}$

AI system quality model combines the previous two (Figure 6).

The proposed quality models of AI, AI platform, and AI system can be extended and detailed due to additional characteristics that consider the specific characteristics of different disciplines. On the other hand, these models are quite extensive and can be simplified under certain circumstances and according to a certain procedure.

### 6.3. Developing of Basic AI and AI System Quality Models

#### 6.3.1. Building of a Basic AI Quality Model $QS_{ChAIb}$

The basic AI quality model is developed to make it more compact and engineer-friendly to evaluate real AI systems. The basic model can be obtained by optimizing it “vertically” and “horizontally”.

The basic AI quality model differs from the original in the following:Vertical optimization is performed by presenting the model in two levels. Sub-characteristics of the third level are considered at the level of metrics of the corresponding characteristics of the second level;The relevant components of the characteristics that are removed or combined can be considered at the level of criteria used for evaluation and weighed accordingly when evaluating the top-level characteristics;The RSP characteristic is removed because it intersects with other characteristics of this level:(a)ETN and LFL—in terms of responsibility for compliance with ethical and legal norms;(b)TST—in terms of responsibility for complying with the requirements of the user as a whole. Moreover, the requirement to inform him in case of possible breach, which is part of the responsibility, can be considered compulsory and therefore considered in the assessment of reliability;(c)EXP—in terms of suitability for verification and provision of information in the event of a breach of the relevant rules and requirements that are part of the characteristics of TRP, VFB;LFL characteristics are combined with ETF, as they are similarly worded and differ only in references to ethical and legal norms, the boundary between which is not always clear. After unification, the characteristics of ETN are formulated as follows: ethics—the ability of AI to comply with current moral standards, laws and regulations regarding the results of surgery;The characteristics of HMA and HMO are combined because they are usually considered together and can complement each other at the metric level. The new definition of HMA is the ability of AI to enable the user to make autonomous informed decisions about the use of AI based on control and to interfere in some way in its functioning;Accountability of ACN and causability CSL causality are combined with transparency TRP as they can be considered as additional transparency metrics. Transparency can then be defined as the ability of AIs to describe, test and repeat models, individual components and algorithms according to which decisions are made, to determine cause-and-effect relationships and to report on performance in a defined form do;Acceptability of ACP is excluded as separate, since it is in fact a “soft” component of the actual credibility, the definition of which does not require adaptation.

Thus, the parenthesis view of the basic AI quality model can be represented as follows:$$QS_{ChAIb}=\left\{\mathrm{ETH}\;\left\{\mathrm{FRN},\;\mathrm{GRS},\;\mathrm{HMA},\;\mathrm{RDR}\right\},\;\mathrm{EXP}\;\left\{\mathrm{CMT},\;\mathrm{CMH},\;\mathrm{TRP},\;\mathrm{INP},\;\mathrm{INR},\;\mathrm{VFB}\right\},\;\mathrm{TST}\;\left\{\mathrm{DVS},\;\mathrm{RSL},\;\mathrm{RBS},\;\mathrm{SFT},\;\mathrm{SCR},\;\mathrm{ACR}\right\}\right\}.$$

This model is also described by a graph (Figure 7), which is a sub-graph of the general $QS_{ChAI}$ model and includes 19 characteristics.

#### 6.3.2. Building of a Basic AI Platform Quality Model $QS_{ChAIPb}$

The optimization of the AI platform quality model was performed according to the same principles and the basic variant thereof was obtained (Figure 8).

Optimization was carried out as follows:One lower level of detail (sub-characteristics: DGV, MTR, TRF, and GRN) is removed;Accessibility of ACS characteristic is absorbed by availability AVL because availability is often seen as synonymous or as a component of readiness. The definition of AVL after this merge does not need to be adapted, as it includes the term “availability”;The IFS feature is absorbed by the maintainability MNT and usability USB characteristics. First, it intersects with maintainability, as the provision and use of information is necessary for the recovery, prevention, and modernization of AI system. Second, the information provided to the user for the convenience of using AI system is an important component of the quality of human-machine interfaces.

#### 6.3.3. Building of a Basic AI System Quality Model $QS_{ChAISb}$

This basic model is a simple combination of basic models $QS_{ChAIb}$ and $QS_{ChAIPb}$ (Figure 9).

## 7. Case Study

Consider examples of building a quality model for real AI systems based on the proposed models. Such models can be used to justify the requirements for the developed systems or to check their implementation and adapt design solutions. The process of building models for real AI systems can be called profile development or requirements profiling. Profiling is implemented by defining quality characteristics at each level of the model hierarchy that is important to the system being analyzed.

This task is solved for the two systems. The basic model of AI system quality was chosen for further consideration and use (Figure 9).

The purpose of the case study is to illustrate how the proposed model can be used to obtain a profile, i.e., models for a particular system with artificial intelligence, considering the characteristics of its (system) use, as well as requirements for its non-functional characteristics. Thus, the solution to this problem is to determine the set of characteristics (formally separating the subgraph from the general graph of the quality model), which must be considered for a particular system (class of systems).

The procedure for obtaining a profile of quality characteristics (requirements) that are important for the system was as follows:Each of the characteristics and sub-characteristics of the model (Figure 9) was analyzed separately;The experts (authors of the article and developers of analyzed systems) determined the need to include the characteristics in the AIS quality model;the decision to include these characteristics in the quality model was made by consensus after considering the opinions of the experts. From our point of view, such a consensus-based approach is acceptable considering the complexity of the tasks and the lack of principal differences in opinions of the experts.

### 7.1. Quality Model of the UAV’s Video Navigation System as an AI System

The first example relates to the UAV’s video navigation system (VNS), which can be used as a separate device or as part of the UAV fleet [116] in monitoring, reconnaissance, etc. The system is based on a convolutional neural network (CNN) with a multilevel structure [117]. The architecture consists of a convolutional neural network for visual features, an extreme learning machine for estimating position bias, and an advanced classifier of extreme information values to predict UAV barriers.

The quality model of VNS as an AI system with built-in CNN is shown in Figure 10. The characteristics which should be considered are marked in gray color. The model features are as follows:At the first level, all three main characteristics of AI quality include: ethics ETH, explainability EXP, and trustworthiness TST;Under the sub-characteristics of ethics the fairness FRN is left, because for VNS the component of FRN definition is important—to reduce the risks of anomalies due to erroneous assumptions and errors in the process of minimizing model setting; for explainability EXP all sub-characteristics are included except for interactivity INR, taking into account the autonomous operation of the UAV; for trustworthiness TST all sub-characteristics are also included, except for diversity DVS, as the application of the multi-versatility principle in on-board systems is limited by the need to minimize overall mass and energy performance;For the AI platform level, all characteristics are included except for the audibility ADT, controllability CNT, and sustainability SST with respect to the purpose and functions of the VNS.

### 7.2. Quality Model of the Decision Support System for the Diagnosis of Urological Diseases as an AI System

The second case illustrates the construction of a quality profile for a decision support system for the diagnosis of urological diseases, based on a distributed three-tier system of neural network modules with additional training [118]. At the first level—the diagnostic urological office—the patient is examined directly and the relevant parameters of urological fluorograms are measured and processed to support the doctor’s decision about the patient’s diagnosis using neural network modules; at the second group level—the regional network of hospitals—the accumulation and exchange of cases of urological diseases is carried out to ensure intensive training of neural network modules of all hospitals; at the third level, the interregional level, information is exchanged between groups for further study, taking into account the details of each region.

The model of the quality of the decision support system for the diagnosis of urological diseases (SDUD) as an AI system with a built-in convolutional neural network is shown in Figure 11. Its features are as follows:At the first level, the model includes all three characteristics (ethics ETH, explainability EXP, and trustworthiness TST);For ethics, all sub-characteristics of fairness FRN are included for obvious reasons due to its definition; for explainability EXP all sub-characteristics are included: fairness FRN, graspability GRS, and human agency HMA except redress RDR, considering the fact that it is a decision support system for diagnosis;Explainability of EXP is represented by all six sub-characteristics as it is important for the medical system;Trustworthiness is represented by four of the six characteristics, except for diversity DVS and safety SFT, given the lack of functions for forming control effects on patients;For the level of the AI platform, only controllability CNT is not included due to the lack of functions for forming control effects and the peculiarities of the purpose of VNS.

Therefore, if we compare the quality models for these systems, we can deduce a more saturated model for the decision support system for the diagnosis of urological diseases as an AI system relative to the basic—its profile contains 23 characteristics and sub-characteristics out of 27, and for VNS—19 out of 27.

Graph of quality model for VNS in fact is a subgraph of graph for SDUD. The graph of SDUD has the following additional characteristics and sub-characteristics which should be considered for assessment of AIS quality: characteristics ADT and SST, GRS, HMA, INR, and SFT.

## 8. Conclusions

### 8.1. Discussion

Despite a large number of publications and the availability of a number of high-level documents issued by reliable national and international institutions, it must be concluded that there is no structured and complete set of AI characteristics, which can be called a quality model similar to an existing and generally accepted software quality models.

The main result of the study is the set of quality models for artificial intelligence, AI platforms, and AI systems. These models are based on the analysis and harmonization of definitions and dependencies of quality traits specific to the AI itself and AI systems. According to our conclusions, some of the characteristics were common to these two entities in the sense that they could be provided at the level of development of the AI itself and its platform.

We tried to select characteristics and build quality models in such a way as to eliminate duplication, ensure completeness of presentation, determine the specific peculiarities of each of the characteristics, and distinguish between the characteristics of the AI and the platform on which AI has been deployed. Clearly, it is extremely difficult to create a model that fully meets such requirements, so the proposed options need to be supplemented and improved considering the rapid development of technologies and applications of AI.

Quality models are presented in this study in various forms—analytical (brackets), tabular, and graph forms, which are convenient for substantive and formal analysis. These models develop the results of [115] and offer the possibility to obtain partial quality profiles for specific developments, considering the details of the respective systems, as demonstrated in the two cases. They can be used as a basis for metric-based AI quality assessment.

The proposed quality models are open and can be supplemented and detailed according to the specific purpose and scope of the AIS. In our opinion, based on the proposed models, it is possible to develop an intersectoral quality standard and requirements for AI (AI systems and AI platforms).

### 8.2. Future Research

Further research needs to be conducted in the following areas:Profiling (addition and detailing) of models for specific industries (healthcare, law, industrial systems, mobility, etc.) considering evolution issues [119]. Such profiling should be accompanied by an overview of the characteristics and sub-characteristics added based on experience in the development and use of AI, for further generalization in the quality models of AI, AI systems, and AI platforms;Development of metrics and algorithms for evaluating AI and AI platforms for each of the proposed characteristics and quality in general. It is advisable to collect and analyze information on various criteria for their inclusion in the general database;Development of tools and case-oriented methods for assessing the quality of AI, AI systems, and AI platforms [120]. They can be based on general assurance case approaches [121,122] as well as functional and cybersecurity assessment approaches [123];Application of internal validation as an additional procedure which can be embedded into AIS assessment [124];Development of content quality models including different aspects of image quality assessment and so on.

## Figures and Tables

**Figure 1 sensors-22-04865-f001:**
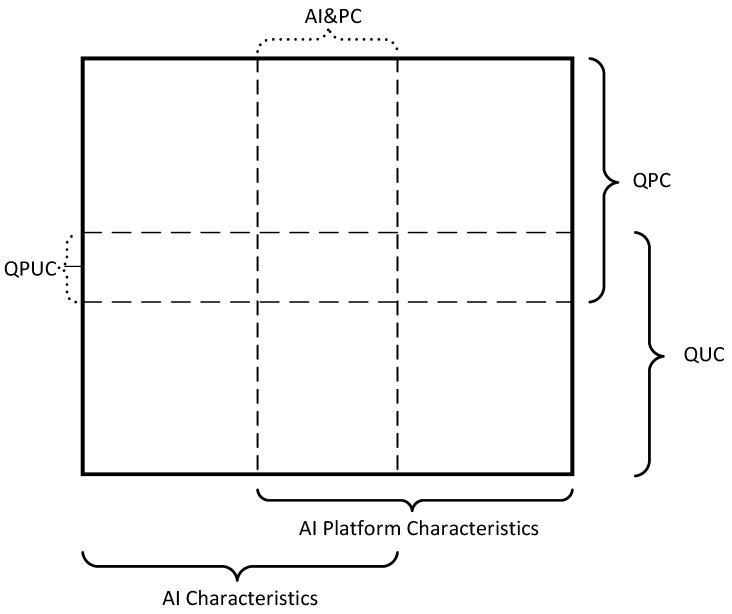
Quality characteristics classification card general view.

**Figure 2 sensors-22-04865-f002:**
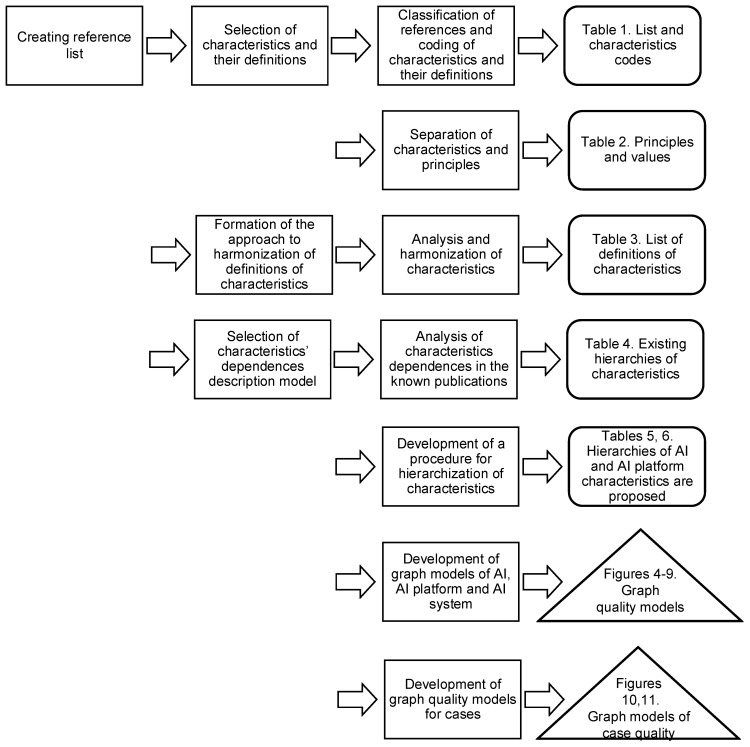
The sequence of AI quality models construction.

**Figure 3 sensors-22-04865-f003:**
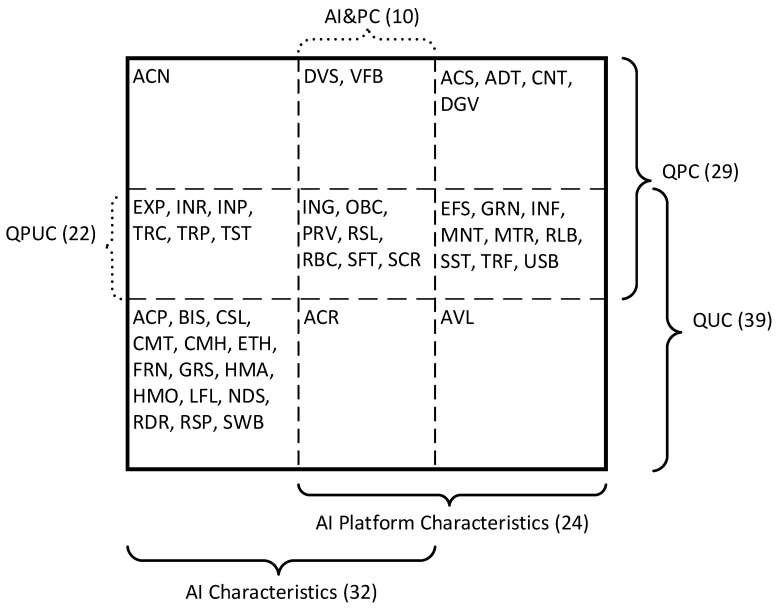
Classification map of AI quality characteristics according to Table 3.

**Figure 4 sensors-22-04865-f004:**
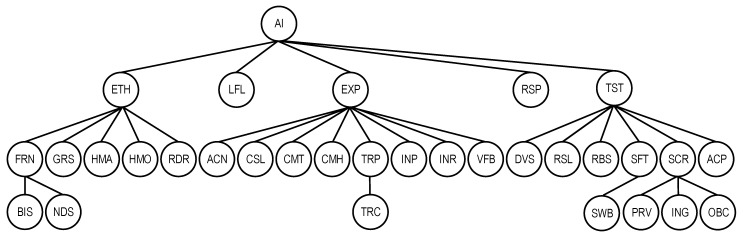
The graph representation of the AI quality model $QS_{ChAI}$.

**Figure 5 sensors-22-04865-f005:**
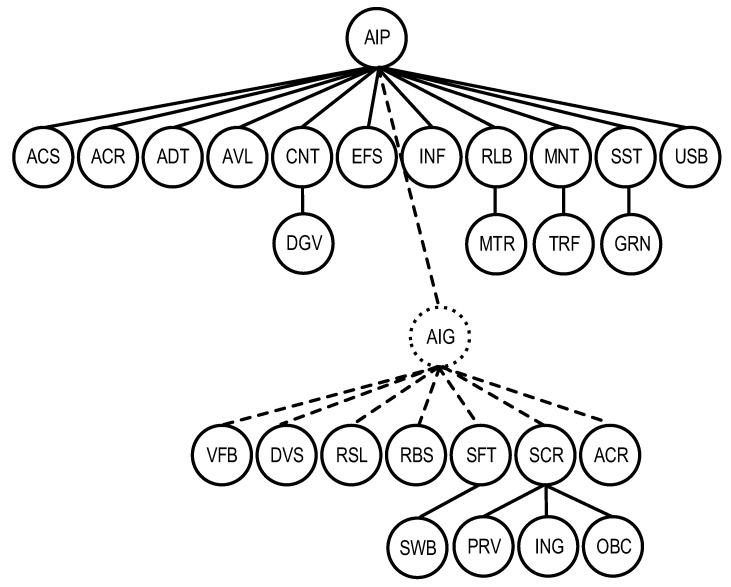
The graph representation of the AI platform quality model $QS_{ChAIP}$.

**Figure 6 sensors-22-04865-f006:**
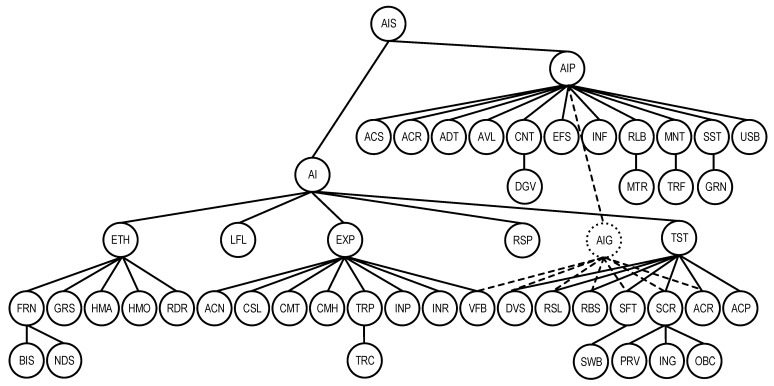
The graph representation of the AI system quality model $QS_{ChAIS}$.

**Figure 7 sensors-22-04865-f007:**
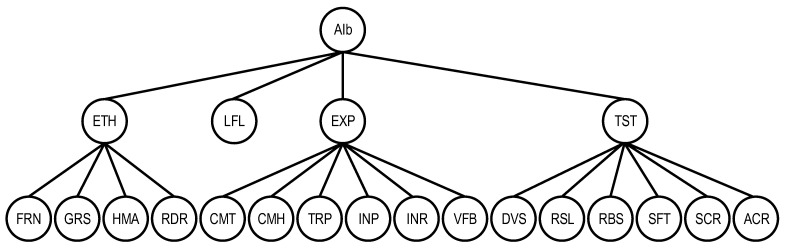
The graph representation of the basic AI quality model $QS_{ChAIb}$.

**Figure 8 sensors-22-04865-f008:**
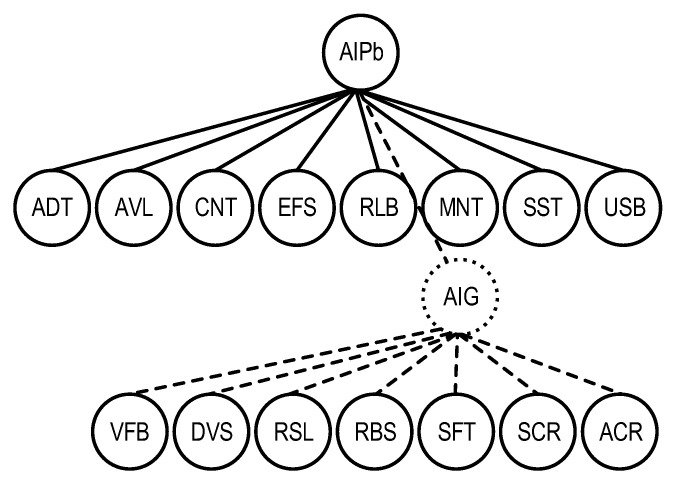
The graph representation of the basic AI platform quality model $\;QS_{ChAIPb}$.

**Figure 9 sensors-22-04865-f009:**
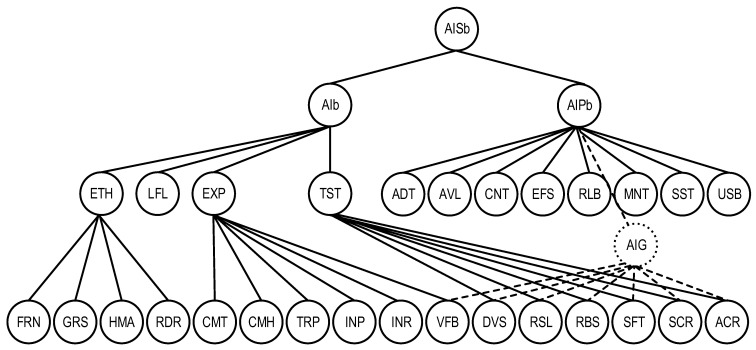
The graph representation of the basic AI system quality model $QS_{ChAISb}$.

**Figure 10 sensors-22-04865-f010:**
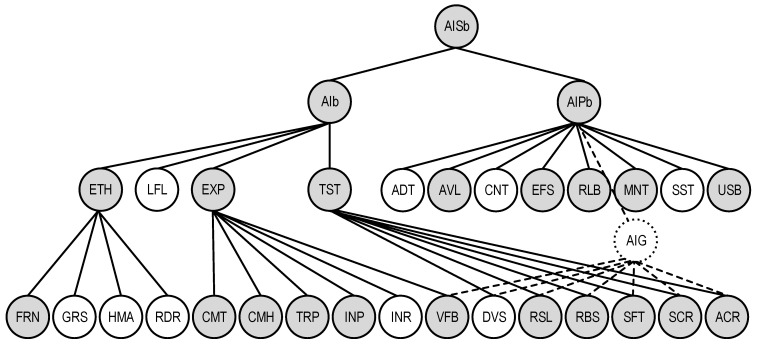
The graph representation of the quality model of VNS as an AI system with built-in CNN.

**Figure 11 sensors-22-04865-f011:**
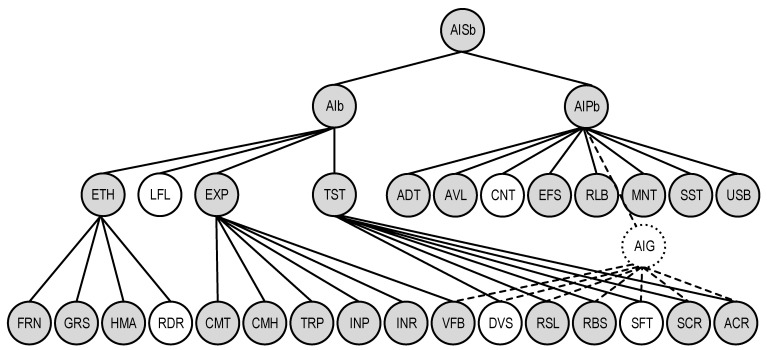
The graph representation of the quality model of the SDUD as an AI system with a built-in convolutional neural network.

**Table 1 sensors-22-04865-t001:** AI characteristics references.

No	Characteristic	Code	References	
			Referenced In	Defined In (First Row—GS, Second Row—GR, Third Row—GV)
**1**	acceptability	ACP	[33,34,35,36,37]	–
				–
				[37]
**2**	accessibility	ACS	[3,31]	[3,31]
				–
				–
**3**	accountability	ACN	[2,3,6,33,38,39,40]	[2,3,6,31,33]
				[40]
				–
**4**	accuracy	ACR	[2,3,17,33]	[2,3,33]
				–
				–
**5**	assurance	ASR	[33]	[33]
				–
				–
**6**	auditability	ADT	[2,3,33,41]	[2,3]
				–
				–
**7**	authenticity	ATH	[6,31]	[31]
				–
				–
**8**	availability	AVL	[6,31]	[31]
				–
				–
**9**	awareness	AWN	[36,38,42]	–
				–
				[42]
**10**	bias	BIS	[2,3,10,15,43,44,45,46,47,48]	[2,3,10,15]
				[44]
				–
**11**	causability	CSL	[49,50,51]	–
				[50]
				–
**12**	completeness	CMT	[31,52]	[31]
				–
				–
**13**	compliance	CML	[33]	–
				–
				–
**14**	comprehensibility	CMH	[36,53,54]	–
				–
				[54]
**15**	communication	CMN	[2,3]	[2,3]
				–
				–
**16**	confidence	CNF	[36,55,56]	–
				–
				[56]
**17**	controllability	CNT	[57]	[57]
				–
				–
**18**	data governance	DGV	[2,3]	[2,3]
				–
				–
**19**	diversity	DVS	[2,3,33]	[2,3,33]
				–
				–
**20**	effectiveness	EFC	[31,58,59]	[31]
				–
				–
**21**	ethics	ETH	[2,3,22,23,60,61,62,63,64]	[2,3]
				–
				–
**22**	environmental well-being	EWB	[2,3]	[2,3]
				–
				–
**23**	exactness	EXC	[65,66,67]	–
				–
				[67]
**24**	explainability	EXP	[1,2,3,13,14,17,24,38,40,46,50,51,52,53,55,58,59,66,68,69,70,71,72,73,74,75,76,77,78,79,80,81,82,83,84,85,86,87,88,89]	[2,3,13]
				–
				–
**25**	explicability	EXL	[2,3]	[2,3]
				–
				–
**26**	fairness	FRN	[2,3,36,40,41,63,90,91,92,93,94]	[2,3]
				–
				–
**27**	fairness and non-discrimination	FND	[22,38]	[22]
				–
				–
**28**	fidelity	FDL	[53,58]	–
				[58]
				–
**29**	fit to purpose	FTP	[33,95]	–
				–
				[95]
**30**	fruitfulness	FRT	[66,96]	–
				–
				[96]
**31**	governance	GVN	[39,97]	–
				–
				[97]
**32**	graspability	GRS	[59]	–
				[59]
				–
**33**	greenness	GRN	[98,99]	–
				–
				[98,99]
**34**	human agency	HMA	[2,3]	[2,3]
				–
				–
**35**	human oversight	HMO	[2,3]	[2,3]
				–
				–
**36**	human oversight and determination	HOD	[22]	[22]
				–
				–
**37**	informativeness	INF	[36,55,100]	–
				–
				[100]
**38**	impartiality	IMP	[39,101]	–
				–
				[101]
**39**	integrity	ING	[6,31,33]	[31]
				–
				–
**40**	intelligibility	INL	[40,53,102]	–
				[102]
				–
**41**	interactivity	INR	[36,38]	–
				[38]
				–
**42**	interpretability	INP	[36,53,73,103]	–
				[103]
				–
**43**	lawfulness	LFL	[2,3]	[2,3]
				–
				–
**44**	literacy	LTR	[38,104]	–
				–
				[104]
**45**	maintainability	MNT	[31,33]	[31]
				–
				–
**46**	maturity	MTR	[31,33]	[31]
				–
				–
**47**	minimization and reporting of negative impacts	MNI	[2,3]	[2,3]
				–
				–
**48**	multi-stakeholder and adaptive governance and collaboration	MGC	[22,38]	[22]
				–
				–
**49**	non-discrimination	NDS	[2,3,22]	[2,3,22]
				–
				–
**50**	objectivity	OBC	[17,105]	[17]
				–
				[105]
**51**	prevention of harm	PRH	[2,3]	[2,3]
				–
				–
**52**	privacy	PRV	[2,3,17,22,31,33,36,38,105]	[2,3,17,22,31]
				[105]
				–
**53**	proportionality and do no harm	PNH	[22]	[22]
				–
				–
**54**	quality	QLT	[6,30,31,33]	[31]
				–
				–
**55**	redress	RDR	[2,3]	[2,3]
				–
				–
**56**	reliability	RLB	[6,17,31,33]	[31]
				–
				–
**57**	resiliency	RSL	[6,17,31,33,39,105]	–
				[105]
				–
**58**	respect for human autonomy	RHA	[2,3]	[2,3]
				–
				–
**59**	responsibility	RSP	[22,31,38,39,40,58]	[22]
				[40]
				–
**60**	robustness	RBS	[2,3,25,58]	[2,3]
				–
				–
**61**	safety	SFT	[2,3,17,22,25,38,41]	[2,3,22]
				–
				–
**62**	similarity	SML	[65,66,106]	–
				–
				[106]
**63**	security	SCR	[6,17,22,25,31,33,38,39]	[22,31]
				–
				–
**64**	societal well-being	SWB	[2,3]	[2,3]
				–
				–
**65**	suitability	STB	[31,107]	–
				–
				[107]
**66**	sustainability	SST	[22,33,38]	[22]
				[108]
				–
**67**	traceability	TRC	[2,3,38,109]	[2,3]
				–
				–
**68**	transferability	TRF	[36,110]	–
				[110]
				–
**69**	transparency	TRP	[2,3,6,25,33,36,38,39,40,55]	[2,3]
				[40]
				–
**70**	trade-offs	TRO	[2,3]	[2,3]
				–
				–
**71**	trustworthiness	TST	[2,3,6,17,25,31,33,36,41,55]	[6,33]
				–
				–
**72**	usability	USB	[6,31,33]	[6,31]
				–
				–
**73**	understandability	UND	[36,57]	–
				[36]
				–
**74**	value proposition	VPR	[33,111]	–
				–
				[111]
**75**	verifiability	VFB	[25,73]	–
				–
				[112]

**Table 2 sensors-22-04865-t002:** Analysis of principles and characteristics of AI and AI systems.

Group of Principles and Values	Reference	Principles	Comment
Ethical principles	[2]	Respect for human autonomy	Can be defined as HMA and HMO
		Prevention of harm	–
		Fairness	Can be defined as FRN
		Explicability	Can be defined as EXP
Values	[22]	Respect, protection, and promotion of human rights and fundamental freedom, and human dignity	–
		Environment and ecosystem flourishing	–
		Ensuring diversity and inclusiveness	–
		Living in peaceful, just, and interconnected societies	–
Principles of activity	[22]	Proportionality and non-infliction of harm	–
		Safety and security	Can be defined as SFT and SCR
		Fairness and non-discrimination	Can be defined as FRN and NDS
		Sustainability	Can be defined as SST
		Right to privacy and data protection	Can be defined as PRV
		Human oversight and determination	Can be defined as HMO, HMA
		Transparency and explainability	Can be defined as TRP, EXP
		Responsibility and accountability	Can be defined as RSP, ECM
		Awareness and literacy	–
		Multi-stakeholder and adaptive governance and collaboration	–

**Table 3 sensors-22-04865-t003:** Analysis of harmonization of AI characteristics definitions.

No	Characteristic	QP/QU	Definition	Way	References
**1**	Acceptability ACP	QU	The ability of the AI to ensure at least partial compliance with customer requirements or consumer expectations	H	[37]
**2**	Accessibility ACS	QP, QU	The ability of the AI platform to enable a user or process that has the appropriate authority to use AI by established rules without waiting longer than a specified or acceptable time	H	[3,31]
**3**	Accountability ACN	QP	The ability of AIs to report in a defined form on the results of operations in a transparent manner	R	[31]
**4**	Accuracy ACR	QU	The ability of the AI and the AI platform to ensure that the results of the requirements and/or functions presented by certain data are close to their true values	H	[2,3,33]
**5**	Auditability ADT	QP	The ability of the AI platform, which is characterized by the degree of suitability for the audit of its ethical and technical components using certain methods and tools	H	[2,3]
**6**	Availability AVL	QU	The ability of the AI platform, which is determined by the degree of its operability, availability, and recoverability	H	[31]
**7**	Bias BIS	QU	It is an AI characteristic that determines the risks of results that are biased due to erroneous assumptions and errors in the process of tuning models (e.g., machine learning)	H	[2,3,10,15]
**8**	Causability CSL	QU	The ability of the AI to determine the cause and effect relationships between events that occur during its use	H	[50]
**9**	Completeness CMT	QU	The ability of the AI to be holistic in terms of compliance with all customer requirements	H	[31]
**10**	Comprehensibility CMH	QU	The ability of the AI to provide the user (or facilitate the user) with an understanding of the explanations sufficient to enable the use of the AI or the information obtained through it to perform other tasks	A	–
**11**	Controllability CNT	QP	The ability of the AI platform to provide opportunities to control and manage the processes of functioning as intended	H	[57]
**12**	Data governance DGV	QP	The ability of the AI platform to provide data control and management capabilities	H	[2,3]
**13**	Diversity DVS	QP	The ability of the AI and AI platforms to minimize the risk of failure to perform specified (defined as necessary) functions or tasks due to failures due to physical and informational factors, using a variety of models, algorithms, and other means	A	–
**14**	Effectiveness EFS	QP, QU	The ability of the AI platform, which is determined by the degree of achievement of the planned results, considering the number of resources spent	H	[31]
**15**	Ethics ETH	QU	The ability of the AI to meet current standards of morality on the results of functioning	H	[2,3]
**16**	Explainability EXP	QP, QU	The ability of the AI to be understood and predictable in terms of purpose and behavior	H	[2,3,13]
**17**	Fairness FRN	QU	The ability of the AI to minimize the risk of biased anomalies in ethical decisions (including lack of favoritism, discrimination on religious, racial, or other grounds, etc.), as well as misconceptions and errors in the modeling process	H	[2,3]
**18**	Graspability GRS	QU	The ability of the AI to provide the user with opportunities for the critical perception of AI in an open and democratic environment	H	[59]
**19**	Greenness GRN	QP, QU	The ability of the AI platform to consume a minimum of energy or other resources, be it energy and/or resource-efficient, and not have an unacceptable impact on the environment	A	–
**20**	Human agency HMA	QU	The ability of the AI to enable the user to make autonomous informed decisions about the use of AI	H	[2,3]
**21**	Human oversight HMO	QU	The ability of the AI to enable the user to control and, if necessary, interfere in a certain way with the functioning of AI	H	[2,3]
**22**	Informativeness INF	QP, QU	The ability of the AI platform to provide the user with useful information to gain new knowledge and generate ideas	H	[100]
**23**	Integrity ING	QP, QU	The ability of the AI and AI platform, which is characterized by the degree of prevention of unauthorized access to modify algorithms or data used by the system	R	[31]
**24**	Interactivity INR	QP, QU	The ability of the AI to provide effective and proactive interaction with the user	H	[38]
**25**	Interpretability INP	QP, QU	The ability of the AI to provide and interpret information in a user-friendly way	H	[103]
**26**	Lawfulness LFL	QU	The ability of the AI to comply with laws and regulations	R	[2,3]
**27**	Maintainability MNT	QP, QU	The ability of the AI platform, which is determined by the degree of efficiency and effectiveness with which it repairs, is maintained and can be modified according to customer requirements (for example through pre- or retraining models)	H	[31]
**28**	Maturity MTR	QP, QU	The ability of the AI platform, which is determined by the degree of compliance with the reliability requirements set by the client	R	[31]
**29**	Non-discrimination NDS	QU	The ability of the AI to enforce ethical standards on non-discrimination on any grounds	H	[2,3,22]
**30**	Objectivity OBC	QP, QU	The ability of the AI and AI platform to prevent the use of compromised or falsified data	R	[105]
**31**	Privacy PRV	QP, QU	The ability of the AI and AI platform to ensure the right to have personal information by user requirements	H	[2,3,105]
**32**	Redress RDR	QU	The ability of the AI to provide available mechanisms to ensure adequate compensation for the effects of adverse effects on humans	H	[2,3]
**33**	Reliability RLB	QP, QU	The ability of the AI platform to perform certain functions for a specified period under certain conditions of use, maintenance, and repair	H	[31]
**34**	Resiliency RSL	QP, QU	The ability of the AI and AI platform to continue to function amid changing requirements, parameters of the physical and information environment, as well as the emergence of unspecified violations and failures	A	–
**35**	Responsibility RSP	QU	The ability of the AI to function considering the expectations of the client (user) by ethical norms, legal regulations, as well as to inform him in case of a possible violation	H	[22,40]
**36**	Robustness RBS	QP, QU	The ability of the AI and AI platform to operate correctly in a wide range of input data and operating conditions and to enter a state of system shutdown if these data and conditions exceed the specified limits	H	[2,3]
**37**	Safety SFT	QP, QU	The ability of the AI and AI platform to avoid the risk of unacceptable damage and loss due to failures due to internal and external causes, and to reduce its consequences with the use of tools built into the AI	H	[2,3,22]
**38**	Security SCR	QP, QU	(Including information and cybersecurity)—the ability of the AI and AI platforms to protect information and physical assets in such a way that other unidentified (unauthorized) persons or systems, including AI and AI platforms, do not have access to them or have such access. such as specified type and level of authorization	H	[22,31]
**39**	Societal well-being SWB	QU	The ability of the AI to take social processes into account and not to harm the physical and mental well-being of people and the well-being of society as a whole	H	[2,3]
**40**	Sustainability SST	QP, QU	The ability of the AI platform to positively influence the sustainable development of economic, social, and environmental environments, as well as to develop without the use of additional resources that are unacceptable in terms of volume or security factors	H	[22,108]
**41**	Traceability TRC	QP, QU	The ability of the AI to track user compliance in a user-friendly way, to search for and document errors and inconsistencies at every stage of the life cycle	H	[2,3]
**42**	Transferability (or interoperability as portability for SW quality stds.) TRF	QP, QU	The ability of the AI platform or software and/or hardware to implement AI, which is determined by the degree of adaptability for the transfer of information, knowledge, and tools to other systems or platforms	H	[111]
**43**	Transparency TRP	QP, QU	The ability of the AI to describe, test, and reproduce models, individual components, and decision-making algorithms	H	[2,3,40]
**44**	Trustworthiness TST	QP, QU	The ability of the AI, which is characterized by the degree of confidence of the user or other stakeholder (developer, auditor, etc.) that the AI meets the requirements and performs its functions in a predictable manner	H	[6,33]
**45**	Usability USB	QP, QU	The ability of the AI platform, which is characterized by the extent to which it can be used by users to achieve specific goals with efficiency, effectiveness, and satisfaction in a given usage context	R	[31]
**46**	Verifiability VFB	QP	The ability of the AI and AI platform, characterized by the degree of suitability for verification by different methods	H	[113]

**Table 4 sensors-22-04865-t004:** Results of the coded references analysis.

Reference	Top-Characteristic	Definition
[2]	Trustworthiness	TST = {HMA, HMO, RBS, SFT, PRV, DGV, TRP {TRC, EXP}, DVS, NDS, FRN, SWB, ACN {ADT, RDR}}
[6]	Trustworthiness	TST = {RLB, AVL, RSL, SCR, PRV, SFT, ACN, TRP, ING, USB}
[17]	Trustworthiness	TST = {ACR, RLB, RSL, OBC, SCR, EXP, SFT, ACN, PRV}
[33]	Trustworthiness	TST = {ACN {RSP, TRP, ADT, MTR}, MNT, ING, SST, ACR, DVS, CML, ACP, USB}
[36]	Explainability	EXP = {CMH, INP, TST, CSL, TRF, INF, FRN, ACS, INR, PRV, TRP}
[38]	Ethics	ETH = {SFT, SCR, SST, PRV, TRP, EXP, RSP, ACN}
[39]	Responsibility	RSP = {ACN, RSL, TRP, SCR}
[40]	Responsibility	RSP = {ACN, TRP, FRN, ETH}
[41]	Trustworthiness	TST = {FRN, EXP, ADT, SFT}
[52]	Explainability	EXP = {CMT, INP}
[53]	Explainability	EXP = {INP{CMH}}
[55]	Explainability	EXP = {TST, INF, TRP}
[57]	Explainability	EXP = {INP, CMH}
[58]	Responsibility	RSP = {RBS, EXP, ETH, EFC}
[106]	Trustworthiness	TST = {TRP, VFB, EXP, SCR, RBS, SFT}

**Table 5 sensors-22-04865-t005:** The tabular representation of the AI quality model $QS_{ChAI}$.

1st Level	2nd Level	3rd Level
Ethics (ETH)	Fairness (FRN)	Bias (BIS)
		Non-discrimination (NDS)
	Graspability (GRS)	–
	Human agency (HMA)	–
	Human oversight (HMO)	–
	Redress (RDR)	–
Explainability (EXP)	Accountability (ACN)	–
	Causability (CSL)	–
	Completeness (CMT)	–
	Comprehensibility (CMH)	–
	Interactivity (INR)	–
	Interpretability (INP)	–
	Transparency (TRP)	Traceability (TRS)
	Verifiability * (VFB)	–
Lawfulness (LFL)	–	–
Responsibility (RSP)	–	–
Trustworthiness (TST)	Acceptability (ACP)	–
	Accuracy * (ACR)	–
	Diversity * (DVS)	–
	Resiliency * (RSL)	–
	Robustness * (RBS)	–
	Security * (SCR)	Integrity (ING)
		Objectivity (OBC)
		Privacy (PRV)
	Safety * (SFT)	Societal well-being (SWB)

**Table 6 sensors-22-04865-t006:** The tabular representation of the AI platform quality model $QS_{ChAIP}$.

1st Level	2nd Level
Accessibility (ACS)	–
Accuracy * (ACR)	–
Auditability (ADT)	–
Availability (AVL)	–
Controllability (CNT)	Data governance (DGV)
Diversity * (DVS)	–
Effectiveness (EFC)	–
Informativeness (INF)	–
Maintainability (MNT)	Transferability (TRF)
Reliability (RLB)	Maturity (MTR)
Resiliency * (RSL)	–
Robustness * (RBS)	–
Security * (SCR)	–
Safety * (SFT)	–
Sustainability (SST)	Greenness (GRN)
Verifiability * (VFB)	–
Usability (USB)	–

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
