# Peer review of "Quality Models for Artificial Intelligence Systems: Characteristic-Based Approach, Development and Application"

_sensors, 2022, doi:10.3390/s22134865_

Round 1

Reviewer 1 Report

In this paper, the authors review quality models for AI/AIS, including characteristic-based approach and application. Some detailed comments can be found as follows:

1. The concept of AI and AIS are mixture. Please clarify this point.

2. It is not necessary to separate the introduction part and the main contributions of this paper should be summarized at the end of this section.

3. As we know that in multimedia areas, the quality is usually used for describing the images/videos, it would be better to distinguish the AI quality from the content quality.

4. Since content delivery is important for AIS, some content quality models are also suggested to be reviewed, including No-reference image quality assessment via transformers, relative ranking, and self-consistency (2D), Dual-stream interactive networks for no-reference stereoscopic image quality assessment (3D), No-reference quality assessment for 360-degree images by analysis of multifrequency information and local-global naturalness (VR), etc.

5. In the experiments, there lacks the comparison results of different graph models.

6. The experimental results are confusing. More analysis about the existing quality models is commended to be included.

7. It is suggested to further improve the presentation. For example, there exist two boxes for Table 5,6 in Figure 2.

Author Response

Dear reviewer, please see the attachment.

After the corrections are made, we'd like to change the title of the paper to the following "Quality Models for Artificial Intelligence Systems: Characteristic-Based Approach, Development and Application”.

Thank you in advance.

Kindest regards, Authors

Reviewer 2 Report

This paper tackles the significant challenge of defining a quality model for AI systems. While, nowadays, quality in AI is still very much evaluated through the predictive performance of models, the authors look at all the overarching characteristics that define the quality of an AI system, including ethical and compliance aspects.

The work therefore allows for an harmonization and standardization of definitions that are scattered around the literature.

My main concern is that some of the proposed characteristics may be of little use in the sense that they may be difficult to calculate objectively, which limits their usefulness. In that sense, they might be regarded more as desiderata (or desired principles) than actual measures that can be calculated to evaluate the quality of an AI system. 

But, nonetheless, the authors' effort is commendable as they provide a very useful list of relevant characteristics/principles. Moreover, the authors also clearly detail the methodology through which they arrived at their list of characteristics and provide the relevant sources. I therefore think that the paper should be published. 

There are however some issues that can be improved:

- There is an issue with Table 4 (some text at the lower end that is starting on the right side of the table).

- Some tables should be revised to make them easier to read. For instance, in Table 2 it is not clear which principle belongs to which group of principles and values. Maybe some horizontal lines defining these groups could help. The same happens in Tables 5 and 6.

- The paper is generally easy to read and to follow. There are, however, some sentences that are confusing to read, most of the cases simply because they are too long. In other cases there are typos or other minor issues (e.g. "their processing is supported to support"). This should not prevent the paper from being published but requires authors to make an additional effort to improve the quality of the paper.

Author Response

(The authors gave the same response as above.)

Reviewer 3 Report

I am really grateful for reviewing this manuscript. In my opinion, this manuscript can be published once some revision is done successfully. This study made a rare attempt to provide quality models for artificial intellidence systems (AIS). I would like to point out that this was an important move. But this study did not develop evaluation methods for AIS quality models. Every model should be developed and validated based on rigorous standards and this study shouldn't be an exception regarding this issue. 

To my best knowledge, this study presented the most comprehensive analysis of the topic, covering 75 characteristics of artificial intelligence (AI) systems based on the most extensive literature review. I would like to point out that this is a great achievement. No matter how comprehensive it is, however, every qualitative study (like this study) has an issue of internal and external validity. Firstly, I would like to suggest the authors to address the issue of internal validity, i.e., whether the authors cross-checked the results among each other and how strong or weak their agreement was on each of the results. Secondly, I would like to suggest the authors to address the issue of external validity. One way to do it is to interview or survey external AI experts as in the section of "Discussion of validity" in a previous study [Siebert, J., Joeckel, L., Heidrich, J. et al. Construction of a quality model for machine learning systems. Software Qual J (2021). https://doi.org/10.1007/s11219-021-09557-y]. I would like to suggest the authors to address this issue as a limitation of the study in case they do not have the time to interview or survey external AI experts. 

Author Response

(The authors gave the same response as above.)
